# Assessment of Phosphine Resistance in Major Stored-Product Insects in Greece Using Two Diagnostic Protocols

**DOI:** 10.3390/insects15100802

**Published:** 2024-10-14

**Authors:** Paraskevi Agrafioti, Efstathios Kaloudis, Dimitrios Kateris, Christos G. Athanassiou

**Affiliations:** 1Institute for Bio-Economy and Agri-Technology (IBO), Centre for Research and Technology—Hellas (CERTH), Dimarchou Georgiadou 118, 38333 Volos, Greece; d.kateris@certh.gr (D.K.); athanassiou@uth.gr (C.G.A.); 2Laboratory of Entomology and Agricultural Zoology, Department of Agriculture, Crop Production and Rural Environment, University of Thessaly, Phytokou Str. Nea Ionia, 38446 Magnesia, Greece; 3Computer Simulation, Genomics and Data Analysis Laboratory, Department of Food Science and Nutrition, School of the Environment, University of the Aegean, Ierou Iochou 10 & Makrygianni, 81400 Lemnos, Greece; stathiskaloudis@aegean.gr

**Keywords:** beetles, fumigation, Greece, phosphine, protocols, resistance

## Abstract

**Simple Summary:**

This study explores the prevalence of phosphine resistance in stored-grain insects. The research, conducted in Greece, examined 53 key insect species and used two assessment protocols, namely, dose–response and CORESTA, to estimate phosphine resistance. The results showed that 13.3% of field populations were resistant, and mortality rates increased with higher phosphine concentrations. According to the dose–response protocol, 37.7% of field populations were found to be resistant, while all populations were susceptible according to the CORESTA protocol. The observed resistance patterns differ from those reported in other regions. The study emphasizes the importance of tailored fumigation strategies considering insect species’ susceptibility to phosphine and recommends best management practices and rotational strategies to develop effective resistance management plans. The results offer valuable insights into the dynamic landscape of phosphine resistance in stored-product insects and suggest potential avenues for further research and control measures.

**Abstract:**

Post-harvest losses due to insect infestation and spoilage by bacteria and molds pose significant challenges to global cereal production. This study investigates the prevalence of resistance to phosphine, a commonly used grain protection agent, in stored-grain insects. The research, conducted in various storage facilities across Greece, examined 53 populations of key stored-product insect species. Two assessment protocols, namely, dose–response (at 50–1000 ppm for 3 days exposure) and CORESTA (at 300 ppm for 6 days), were used herein to estimate phosphine resistance. The results showed that 13.3% of field populations were resistant, and mortality rates increased with higher phosphine concentrations. Specifically, according to the dose–response protocol, among the 53 field populations, 37.7% were found to be resistant to phosphine, namely, two populations of *O. surinamensis*, one of *S. oryzae*, seven of *T. confusum*, one of *C. ferrugineus*, one of *T. castaneum*, and all populations of *R. dominica*, whereas, according to the CORESTA protocol, all populations were found to be susceptible to phosphine. The observed resistance patterns differ from those reported in other regions of the world. The study highlights the importance of tailored fumigation strategies, considering insect species varying susceptibility to phosphine. It recommends the use of best management practices and rotational strategies, such as combining phosphine with other methods, to develop effective resistance management plans. The results provide valuable insights into the dynamic landscape of phosphine resistance in stored-product insects and suggest potential avenues for further research and control measures.

## 1. Introduction

Post-harvest losses due to insect infestation of stored products and spoilage by bacteria and molds account for 20% and 10% of total cereal production in developing and developed countries, respectively [1,2]. It is estimated that between 60% and 90% of produce is stored for months at a time in various storage facilities such as warehouses, silos, containers, and large bags. More than 600 species of beetles, 70 species of moths, and 355 other arthropod species (specifically, mites) are listed as common insects causing serious losses of various stored agricultural commodities [3]. As a result, stored-product insects are responsible for a significant proportion of quantitative and qualitative losses of stored commodities. Therefore, there is an urgent need to control stored-product insect infestations using effective methods to protect various stored agricultural commodities.

Grain protectants have been widely used to control stored-product insects and continue to play a significant role in protecting grain during storage [4,5,6]. The control of stored-product insects is based on insecticides such as pyrethroids, organophosphates, and fumigants [1,4]. Phosphine (PH_3_) is the most widely used insecticide for controlling stored-product insects in a variety of storage structures, such as horizontal warehouses, containers, silos, tarpaulins, ship holds, and different processing facilities [7,8,9]. It can be applied to a wide range of products and foods such as cereal grains, tobacco, dried fruits, and other raw or processed foods [10,11]. Its ease of use, low application cost, and global acceptance as a residue-free treatment have made this gas incredibly popular in various storage facilities [12,13].

Phosphine has been proven to be effective against stored-product insects and mites [12,14,15]. However, the continued use of phosphine has led to the development of resistance in many pests around the world [16,17,18,19]. Many published data focus on the development of resistance to phosphine on continents around the world, such as Africa, Australia, North America, and Europe [18,19,20,21,22,23,24]. In fact, many species, such as the lesser grain borer, *Rhyzopertha dominica* (F.) (Coleoptera: Bostrychidae), and the rusty grain beetle, *Cryptolestes ferrugineus* (Stephens) (Coleoptera: Laemophloeidae), have developed considerable resistance and can survive at doses much higher than the recommended concentration [17,25,26]. Moreover, the rice weevil, *Sitophilus oryzae* (L.) (Coleoptera: Curculionidae); the maize weevil, *Sitophilus zeamais* (Motschulsky) (Coleoptera: Curculionidae); the red flour beetle, *Tribolium castaneum* (Herbst) (Coleoptera: Tenebrionidae); and the confused flour beetle, *Tribolium confusum* Jacquelin Du Val (Coleoptera: Tenebrionidae), were found to be resistant by many research groups [21,27,28]. Phosphine resistance is very common worldwide, with eggs and pupae being the most resistant of life stages for the majority of species tested. In this regard, severe and persistent losses following phosphine applications have been reported from several areas, including Australia, China, and the USA [14].

The main diagnostic protocols used by many research groups around the world to determine resistance in major storage pests have repeatedly produced inconsistent and sometimes contradictory results [14,17,29]. In principle, the most widely used protocol is the Food and Agriculture Organization (FAO) protocol, which is based on relatively short exposure intervals (from hours to days). This protocol has been modified by several researchers in the world [14,18,19,29,30], making it difficult to compare the results obtained by different research groups. In this case, insects are exposed to concentrations of 30–50 ppm of phosphine for 20 h. Another approach is the dose–response protocol, which is based on different concentrations ranging from 50 to 2000 ppm [14,19,24]. On the other hand, the organization for scientific research on tobacco, the Cooperation Center for Scientific Research Relative to Tobacco (known as CORESTA), has developed a protocol that is based on higher concentrations and longer exposure intervals for the cigarette or tobacco beetle, *Lasioderma serricorne* (F.) (Coleoptera: Anobiidae), and the tobacco moth, *Ephestia elutella* (Hübner) (Lepidoptera: Pyralidae) [31], to compensate for and mitigate failures due to phosphine resistance. Based on the protocols developed by CORESTA for this purpose, the evaluation of beetle resistance is carried out using exposures that last for several days (4–12 days), at phosphine concentrations usually ranging from 200 to 700 ppm. Depending on the temperature, in the case of *L. serricorne*, the CORESTA protocol suggests using 200 ppm for 4 days for the susceptible populations at >20 °C, whereas at 16–20 °C, a concentration of 300 ppm is applied for 6 days [31]. For example, Sakka and Athanassiou [19] studied different populations of *L. serricorne* and found that some of the populations tested were able to survive at 200 ppm for 4 days.

Therefore, we evaluated the major stored-product insect species to study the resistance of insect populations from different parts of Greece. This procedure was carried out using both the dose–response and CORESTA protocols for all the populations tested.

## 2. Materials and Methods

### 2.1. Tested Species

Fifty-three populations were sampled from different storage facilities (warehouses, silos, and containers) in Greece (Table 1). Most samples were collected from warehouses (66%), followed by silos (23%) and containers (11%). Samples (at least 100 g) of infested commodities (different cereals) were transferred and reared at the Laboratory of Entomology and Agricultural Zoology, Department of Agriculture Crop Production and Rural Environment, University of Thessaly. The samples were examined for the presence of insects, and the species identified were separated into jars to start rearing the species in a favorable culture medium. The species found were the granary weevil, *Sitophilus granarius* (L.) (Coleoptera: Curculionidae); *O. surinamensis*; *S. oryzae*; *T. confusum*; *C. ferrugineus*; *T. castaneum*; and *R. dominica*.

In all experiments, the laboratory reference of each insect species was included as a “control” and had been kept under laboratory conditions (without exposure to phosphine) for more than 20 years. All rearing was carried out in glass jars containing different substrates for each species. Further information on the geographical regions, insect rearing, and insect substrates was previously presented by Agrafioti et al. [18]. All reared insects were kept in incubation chambers set at 25 °C, 55% relative humidity (r.h.), and continuous darkness. The adult stage (less than 1 month old) was used in the experiments.

### 2.2. Dose–Response Protocol

This protocol uses a modified FAO protocol, also known as a dose–response protocol, in which 20 adults of the species and samples tested (separate species and samples each time) were placed in a 1 L glass jar and exposed to phosphine at concentrations of 50, 100, 200, and 1000 ppm for a 3-day exposure period. At the end of this period, dead (no visible movement) and living (normal movement) insects were recorded and checked under the stereoscope using a paintbrush. All adults were transferred to petri dishes with a small amount of food source in each dish, i.e., cracked wheat (0.5 ± 0.1 g/dish) for *R. dominica*, *S. granarius*, and *S. oryzae*; wheat flour (1.0 ± 0.1 g/dish) for *T. confusum*, *T. castaneum*, and *C. ferrugineus*; and oat flakes (1.0 ± 0.1 g/dish) for *O. surinamensis*, as mentioned by Lampiri et al. [32]. The delayed effect was counted after a 7-day post-exposure period, and the mortality data were recorded.

### 2.3. CORESTA Protocol

In this protocol, 20 adults of the tested species and samples (separate species and samples each time) were placed in a 1 L glass jar and exposed to phosphine at concentration of 300 ppm for a period of 6 days. At the end of the exposure, dead and living individuals were recorded and checked under a stereoscope using a paintbrush. For both protocols, in parallel with our experiments, extra glass jars were used without the addition of phosphine.

For both protocols, the whole experiment was repeated 3 times (by 3 sub-replicates), with new phosphine production each time. Phosphine production was described in detail by Agrafioti et al. [18]. The phosphine concentration was measured by quantitative gas chromatography (GC) using a GC-2010 Plus (Shimadzu, Kyoto, Japan) instrument equipped with a GS-Qcolumn (30 m long × 0.25 mm i.d., 0.25 μm film thickness; MEGA S.R.L., Milan, Italy) as suggested by Sakka and Athanassiou [24]. Additionally, glass tubes from the Draeger Company (Draeger Safety AG & Co., Lubeck, Germany) were used. The protocol was carried out under laboratory conditions set at 25 °C and 55% r.h. in incubator chambers (ELVEM, Attica, Greece, CL1400, dimensions: 210 × 168 × 90 cm).

### 2.4. Data Analysis

For both protocols, the data were analyzed separately for each species and population, by using probit analysis to estimate the lethal concentration values (LC_50_ and LC_99_) based on the insects that did not move. For this purpose, regression analysis was performed using SPSS (IBM SPSS v.26). Afterwards, the mean number of dead individuals and the standard error values were estimated. For the purpose of this analysis, one-way ANOVA was performed to determine whether there were differences among the concentrations tested. The means were separated by Tukey’s HSD (honestly significant difference) test at 0.05.

## 3. Results

### 3.1. Dose–Response Protocol

Based on this protocol, all laboratory populations tested were classified as susceptible after 3 days of exposure at 50, 100, 200, and 1000 ppm (Table 1 and Table 2). Of the 53 field populations, 37.7% were found to be resistant to phosphine: two populations of *O. surinamensis*, one of *S. oryzae*, seven of *T. confusum*, one of *C. ferrugineus*, one of *T. castaneum*, and all populations of *R. dominica* (Table 2). The diagnosis was made according to the FAO protocol, and populations with LC_50_ values <2.85 ppm were characterized as resistant, as suggested by Wakil et al. [33]. Among the *T. confusum* field populations, GA12 was the most resistant, with an LC_50_ of 43 ppm, i.e., it was 61.42 times more resistant than the susceptible field population (Table 2). The highest LC_50_ and LC_99_ values were observed for *R. dominica* GA3 (308.6 ppm) and *R. dominica* ASC11 (1766.4 ppm), respectively (Table 2). As the concentration increased from 50 ppm to 1000 ppm, the survival rate gradually decreased (Table 3). For example, when adults were exposed to 50, 100, 200, and 1000 ppm for 3 days, the survival rates of the 53 field populations were 73.5, 58.4, 41.5, and 0%, respectively (Table 3). In addition, significant differences were found between the concentrations tested in some field populations (Table 3).

Regarding the post-exposure period, no surviving individuals were observed in the laboratory populations (Table 4 and Table 5). Of the 53 field populations, 43.3% were found to be resistant to phosphine, which was 5.6% higher than the resistance rate at 3 days of exposure (immediate effect). Specifically, *T. confusum* was found to be resistant in 11 of these 27 populations (Table 4). Among *S. oryzae* field populations, ASC11 was the most resistant, with an LC_50_ value at 34.2 ppm, i.e., it was 48.85 more resistant than the susceptible field population, whereas among the *T. confusum* populations, GA12 was the most resistant, with an LC_50_ value at 47.5 ppm, i.e., it was 158.33 more resistant than the most susceptible field population (Table 4). The highest LC_50_ and LC_99_ values were observed for *R. dominica* ASC14 (209.1 ppm) and *R. dominica* ASC11 (2758 ppm), respectively (Table 4). In addition, 36, 26, 15, and 0 populations were found to have surviving adults at 50, 100, 200, and 1000 ppm, respectively, after a 7-day exposure interval (Table 5). Significant differences were also found between the concentrations tested (Table 5).

### 3.2. CORESTA Protocol

Based on this protocol, all laboratory populations were found to be susceptible to 6 days of exposure at 300 ppm. Similarly, all adults of the field populations were dead in both the immediate and post-exposure evaluation periods. All field populations were characterized as susceptible.

Figure 1 presents a map giving a comparative overview of phosphine resistance in stored-product insects in different regions of Greece, focusing on the efficiency of two evaluation protocols: dose–response and CORESTA. The bars highlight the number of insect species/populations within each region with LC_99_ values below 300 ppm (favoring the dose–response protocol) or above 300 ppm (indicating potential efficacy with the CORESTA protocol).

For example, in the Peloponnese region, the dose–response protocol appears more effective, with two species/populations demonstrating LC_99_ values below 300 ppm, compared to five species/populations with LC_99_ values above 300 ppm under the CORESTA protocol. Conversely, in Sterea Hellas, the dose–response protocol shows promise, with three species/populations below 300 ppm, while the CORESTA protocol indicates efficacy for two species/populations. Thessaly and Macedonia also exhibit variations, highlighting the importance of considering regional differences in insect susceptibility to phosphine.

## 4. Discussion

To the best of our knowledge, this is the first work in which the phosphine resistance of several Greek stored-product insects has been estimated and characterized using these two evaluation protocols, dose–response and CORESTA. Based on our results reported here, 13.3% of the field populations were characterized as resistant to phosphine, i.e., 20 out of a total of 53 field populations collected. The mortality rate increased steadily with increasing phosphine concentration. This should be taken into account by storage operators to avoid fumigation failures, such as individuals surviving after fumigation has been completed.

The present results are strikingly different from those found in other areas of the world [14,29,30]. The levels of phosphine resistance in *S. granarius*, *S. oryzae*, *T. confusum*, *C. ferrugineus*, *T. castaneum*, and *R. dominica* reported in the present study are based on a 3-day fumigation period (dose–response) using the adult stage only. In *O. surinamensis*, for example, two populations were found to be resistant to phosphine, and the LC_99_ values reached 557.8 and 152.1 ppm for ASC11 and GA1, respectively. In contrast, Gautam et al. [23] found that *O. surinamensis* OKWat population originating from Oklahoma required 2641.1 ppm. In *T. castaneum* populations, Gautam et al. [29] reported that the most resistant population was 49 times more resistant than the susceptible one, in which 356.9 ppm was required to kill 99.9% of the total population, whereas in our case, one to five populations were characterized as resistant with an LC_99_ value of 290.4 ppm. Reports of phosphine resistance in *R. dominica* have been documented by Lorini et al. [26], Aulicky et al. [21], Chen et al. [34], and Afful et al. [22] and indicated that this species has the highest incidence of resistance. Similarly, in our case, all *R. dominica* populations tested were classified as resistant to phosphine, as most of them required more than 1000 ppm to kill the individuals. Additionally, in Australia and Turkey [17,35], the very high frequency of strong resistance in *C. ferrugineus* was highlighted, whereas in Greece, this phenomenon has not been reported in the same species but only in *R. dominica*, a very common species in Greek storage facilities [36,37]. In *S. oryzae* and *T. confusum* populations, less than 300 ppm was needed, which was extremely low compared to populations from China, India, and Vietnam [27,38].

In this work, two protocols were assessed against stored-product insect species: the dose–response protocol, based on exposing the tested adults to 50, 100, 200, and 1000 ppm phosphine for 3 days and on the CORESTA protocol, which uses 300 ppm for 6 days (depending on temperature). Despite the fact that the protocols tested here were completely different, there were some similarities in the mortality data. The dose–response protocol has been regarded as an important diagnostic protocol for resistance to phosphine [29,39]. Based on our results, in 31 out of 53 populations were found to be susceptible at 200 ppm for 3 days, as mortality was complete (100%), whereas exposure at 1000 ppm for the same exposure interval almost all populations were dead. Specifically, at 200 ppm, in 10 out of 22 resistant populations, the survival rates were <5%, meaning that almost all individuals were close to death, which means that the entire populations were affected. At the lowest exposure concentration, 50 ppm, a high number of individuals survived. Contrary results were found in the reference Gourgouta and Athanassiou [40], where 100% mortality was reported for the lesser mealworm, *Alphitobius diaperinus* Panzer (Coleoptera: Tenebrinionidae), and the yellow mealworm, *Tenebrio molitor* L. (Coleoptera: Tenebrionidae), even at the lowest concentration of 50 ppm for 3 days of exposure. The CORESTA [31] makes two recommendations for effective phosphine treatment against *L. serricorne*. The tobacco temperature must be checked before starting the fumigation. Depending on the commodity temperature, the phosphine concentration ranges from 200 to 300 ppm for 4 to 6 days, respectively [31]. There have been populations that, when exposed to either “normal” or “high-dose” CORESTA protocol exposures, had surviving individuals recorded [19,41]. However, in our case, this is the first study in which all stored-product species were classified as susceptible to phosphine, as complete mortality was observed. This protocol is the most rigorous of all protocols because it has the longest duration of all protocols used by many research groups around the world, highlighting the importance of duration in commercial facilities (i.e., 6 days of exposure).

Our screening showed that there are certain insect species that are more likely to be characterized as “weakly” resistant, such as *R. dominica*, which may be related to the type of commodity. For example, resistance was more frequently detected in insects found in raw grain (i.e., in warehouses and silos) than in insects found in flour and related processed products, as grain is repeatedly treated with phosphine (unlike processed products) [18]. In our case, most of the selected samples came from warehouses. Based on “real-world” phosphine fumigations, high survival rates of exposed adults and significant numbers of offspring were observed in warehouses and silos, with the exception of container fumigations, where 100% mortality was observed [42,43,44]. This was partly due to the short duration of fumigation combined with the low phosphine concentrations tested in the warehouses. It should be noted that some storage facilities, such as warehouses and silos, are not designed for phosphine fumigations, and it is not uncommon for “fumigation failures” to be confused for the presence of resistance.

The results of the present study show the importance of exposure time in combination with gas concentration. The data show that regardless of the concentration level, the exposure interval is probably more critical than the gas concentration for insect mortality [21]. As the exposure time increased, all individuals died, which means that they were susceptible to phosphine, whereas at shorter exposure intervals, i.e., 3 days, nonzero survival rates were observed. In the fumigation trials, complete parental mortality was observed after 3.5 days of exposure, and offspring were recorded 2 months later, as mentioned by Sotiroudas et al. [42] and Agrafioti et al. [43,44]. Moreover, it should be noted that the CORESTA protocol can be successfully applied not only to the tobacco moth and the tobacco beetle but also to the major stored-product beetle species with success. Furthermore, Agrafioti et al. [43,44] suggested that effective fumigations would result in complete parental mortality and complete suppression of progeny production for all populations tested.

## 5. Conclusions

Best management practices for fumigation, as well as the combined use of phosphine and other methods in a rotational strategy, can provide the necessary information to develop a reliable plan of action to control this phenomenon. For example, a modified atmosphere, which can be applied in either chambers or silos, can be effective with a very short exposure interval, which is an advantage of using this method against stored-product insects with varying susceptibility to phosphine [45,46,47,48].

## Figures and Tables

**Figure 1 insects-15-00802-f001:**
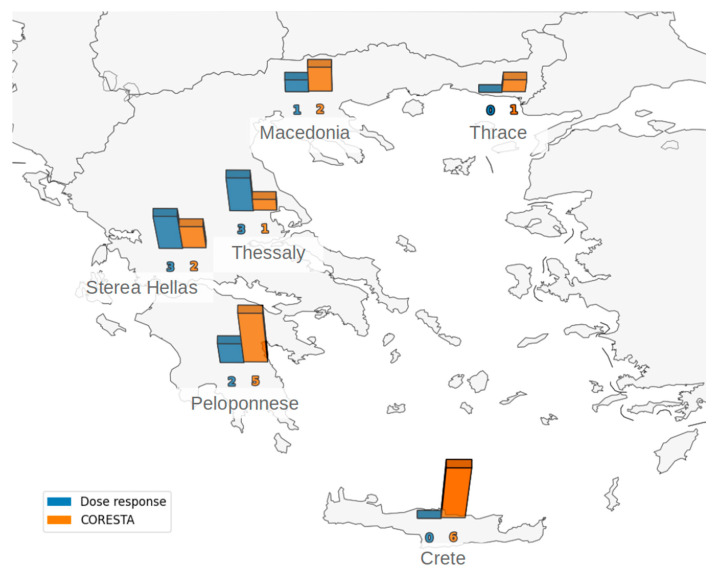
Regional comparison of phosphine resistance in stored-product insects: LC_99_ analysis between the dose–response (>300 ppm) and CORESTA (<300 ppm) protocols in Greek storage facilities.

**Table 1 insects-15-00802-t001:** Insect species sampled from different substrates located in different geographic regions in Greece.

Population Code	Area Sampled	Commodity Sampled	Species Found
ASC3	Sterea Hellas	Wheat flour	*T. confusum*
ASC7	Thrace	Wheat flour	*S. oryzae*
ASC8	Thrace	Wheat flour	*S. granarius*, *C. ferrugineus*
ASC9	Thessaly	Wheat flour	*T. confusum*
ASC10	Thessaly	Wheat flour	*T. castaneum*, *O. surinamensis*
ASC11	Peloponnese	Wheat	*T. castaneum*, *R. dominica*, *O. surianemensis*, *S. oryzae*
ASC14	Sterea Hellas	Cereals	*T. confusum*, *R. dominica*
ASC15	Peloponnese	Semolina	*T. castaneum*
ASC16	Peloponnese	Wheat flour	*T. confusum*
ASC19	Thessaly	Wheat flour	*T. castaneum*
GA1	Peloponnese	Βarley flour	*S. oryzae*, *O. surinamensis*
GA2	Macedonia	Wheat	*R. dominica*, *C. ferrugineus*, *S. oryzae*
GA3	Thessaly	Wheat	*R. dominica*
GA6	Thessaly	Wheat	*R. dominica*
GA12	Thessaly	Wheat	*T. confusum*, *R. dominica*
U1	Macedonia	Wheat	*S. oryzae*
V1	Crete	Wheat flour	*T. confusum*
V2	Crete	Wheat flour	*T. confusum*
P1	Crete	Bran flour	*T. confusum*
PP1	Crete	Residues	*T. confusum*
P2(1)	Crete	Residues	*T. confusum*
P2(2)	Crete	Residues	*T. confusum*
P3	Crete	Residues	*T. confusum*
P4	Crete	Residues	*T. confusum*
F1	Crete	Residues	*T. confusum*
F2	Crete	Residues	*T. confusum*
F3	Crete	Residues	*T. confusum*
F4	Crete	Residues	*T. confusum*
F5	Crete	Residues	*T. confusum*
S1	Crete	Residues	*T. confusum*
S2	Crete	Residues	*T. confusum*
S3	Crete	Residues	*T. confusum*
SK1	Crete	Residues	*T. confusum*
SK2	Crete	Residues	*T. confusum*
SKG1	Crete	Residues	*T. confusum*
SKG2	Crete	Residues	*T. confusum*
AGR-01	Sterea Hellas	Grain byproducts	*S. oryzae*, *R. dominica*, *T. confusum*
AGR-03	Sterea Hellas	Rice	*R. dominica*
AGR-04	Sterea Hellas	Barley flour	*T. castaneum*
AGR-05	Sterea Hellas	Wheat flour	*T. confusum*
EXT1	Macedonia	Rice	*O. surinamensis*
LAB	University of Thessaly	Preferred rearing media	*S. oryzae*, *S. granarius*, *O. surinamensis*, *T. confusum*, *T. castaneum*, *R. dominica*, *C. ferrugineus*

**Table 2 insects-15-00802-t002:** LC_50_ and LC_99_ (with the confidence intervals) for the mortality response in adults after exposure to 50, 100, 200, and 1000 ppm for 3 days (in all cases, *df* = 34), determined using probit analysis.

Species	Populations	LC_50_ (Confidence Interval)	LC_99_ (Confidence Interval)	Y-Intercept	x^2^	*p*	Diagnosis ^a^
*S. granarius*	ASC8	**	**	**	**	**	Susceptible
	LAB	**	**	**	**	**	Susceptible
*O. surinamensis*	ASC10	**	**	**	**	**	Susceptible
	ASC11	102.5 *	557.8 *	−6.35	400.0	<0.01	Resistant
	GA1	72.9 (46–107)	152.1 (104.8–1049)	−13.58	954.0	<0.01	Resistant
	EXTR1	**	**	**	**	**	Susceptible
	LAB	**	**	**	**	**	Susceptible
*S. oryzae*	GA2	**	**	**	**	**	Susceptible
	GA1	0.7 *	27.6 *	0.16	13.8	0.99	Susceptible
	ASC7	**	**	**	**	**	Susceptible
	ASC11	43 (26.7–54.3)	231.6 (156.8–585.3)	−5.19	124.9	<0.01	Resistant
	U1	7.0 *	58.8 *	−2.15	13.1	1.00	Susceptible
	AGR-01	**	**	**	**	**	Susceptible
	LAB	**	**	**	**	**	Susceptible
*T. confusum*	ASC3	12.2 *	73 *	−3.27	50.0	0.03	Resistant
	ASC9	**	**	**	**	**	Susceptible
	ASC14	20.1 (3–30)	86.5 (69.3–214)	−4.79	30.5	0.63	Resistant
	V1	17.2 *	136.1 *	−3.21	102.5	<0.01	Resistant
	V2	13.9 *	75 *	−3.63	18.8	0.98	Resistant
	P1	19.2 *	85.1 *	−4.63	24.6	0.88	Resistant
	AGR-01	**	**	**	**	**	Susceptible
	AGR-05	**	**	**	**	**	Susceptible
	ASC16	0.8	27.6 *	0.16	13.8	0.99	Susceptible
	GA12	43.0 *	177.2 *	−6.17	123.1	<0.01	Resistant
	F1	**	**	**	**	**	Susceptible
	F2	**	**	**	**	**	Susceptible
	F3	**	**	**	**	**	Susceptible
	F4	0.7	27.6	0.16	14.0	0.99	Susceptible
	F5	**	**	**	**	**	Susceptible
	PP1	**	**	**	**	**	Susceptible
	P2(1)	**	**	**	**	**	Susceptible
	P2(2)	**	**	**	**	**	Susceptible
	P3	**	**	**	**	**	Susceptible
	P4	**	**	**	**	**	Susceptible
	S1	0.7	27.6	0.16	13.7	0.99	Susceptible
	S2	**	**	**	**	**	Susceptible
	S3	**	**	**	**	**	Susceptible
	SKG1	9.0	64.6	−2.61	15.1	0.99	Resistant
	SKG2	**	**	**	**	**	Susceptible
	SK1	**	**	**	**	**	Susceptible
	SK2	**	**	**	**	**	Susceptible
	LAB	**	**	**	**	**	Susceptible
*C. ferrugineus*	ASC8	1.8 *	42.3 *	−0.45	25.8	0.84	Susceptible
	GA2	16.5 *	375.6 *	−2.09	56.3	0.01	Resistant
	LAB	**	**	**	**	**	Susceptible
*T. castaneum*	ASC10	**	**	**	**	**	Susceptible
	ASC11	63.0 *	290.4 *	−6.31	119.6	<0.01	Resistant
	ASC15	**	**	**	**	**	Susceptible
	ASC19	**	**	**	**	**	Susceptible
	AGR-04	**	**	**	**	**	Susceptible
	LAB	**	**	**	**	**	Susceptible
*R. dominica*	ASC11	219.3 *	1766.4 *	−6.02	93.0	<0.01	Resistant
	GA6	297.9 *	1134.6 *	−9.91	317.9	<0.01	Resistant
	GA3	308.6 *	1541 *	−8.29	236.2	<0.01	Resistant
	GA2	18.8 *	141.7 *	−3.38	45.8	0.08	Resistant
	AGR-03	264 *	1296.9 *	−8.14	101.7	<0.01	Resistant
	GA12	156.7 *	309.5 *	−17.27	2191	<0.01	Resistant
	ASC14	281.6 *	1668.3 *	−7.37	150.0	<0.01	Resistant
	AGR-01	23.0 *	303 *	−2.83	157.9	<0.01	Resistant
	LAB	**	**	**	**	**	Susceptible

* The confidence interval could not be estimated precisely. ** The LC value and the resistance ratio could not be estimated precisely. ^a^ The diagnosis reported here for the dose–response protocol was the same as the diagnosis determined by FAO protocol, with resistance defined by an LC_50_ value of >2.85 as suggested by Wakil et al. [33].

**Table 3 insects-15-00802-t003:** Mean numbers (% ± SE) of adults of *S. granarius*, *O. surinamensis*, *S. oryzae*, *T. confusum*, *C. ferrugineus*, *T. castaneum*, and *R. dominica* that were found dead at 50, 100, 200, and 1000 ppm after 3 days of exposure (in all cases, *df_total_* = 35).

Species	Population	50	100	200	1000	F	*p*
*S. granarius*	ASC8	100.0 ± 0.0	100.0 ± 0.0	100.0 ± 0.0	100.0 ± 0.0		
	LAB	100.0 ± 0.0	100.0 ± 0.0	100.0 ± 0.0	100.0 ± 0.0		
*O. surinamensis*	ASC10	100.0 ± 0.0	100.0 ± 0.0	100.0 ± 0.0	100.0 ± 0.0		
	ASC11	4.4 ± 1.9 A	28.3 ± 14.0 A	22.2 ± 12.1 A	100.0 ± 0.0 B	20.368	<0.001
	GA1	6.6 ± 2.5 A	70.0 ± 10.1 B	41.1 ± 12.2 C	100.0 ± 0.0 B D	24.542	<0.001
	EXTR1	100.0 ± 0.0	100.0 ± 0.0	100.0 ± 0.0	100.0 ± 0.0		
	LAB	100.0 ± 0.0	100.0 ± 0.0	100.0 ± 0.0	100.0 ± 0.0		
*S. oryzae*	GA2	100.0 ± 0.0	100.0 ± 0.0	100.0 ± 0.0	100.0 ± 0.0		
	GA1	99.5 ± 0.5	100.0 ± 0.0	100.0 ± 0.0	100.0 ± 0.0	1.00	0.405
	ASC7	100.0 ± 0.0	100.0 ± 0.0	100.0 ± 0.0	100.0 ± 0.0		
	ASC11	56.5 ± 11.02 A	63.9 ± 4.3 AB	81.6 ± 3.8 BC	100.0 ± 0.0 CD	9.720	<0.001
	U1	96.5 ± 2.7	99.4 ± 0.6	100.0 ± 0.0	100.0 ± 0.0	1.282	0.297
	AGR-01	100.0 ± 0.0	100.0 ± 0.0	100.0 ± 0.0	100.0 ± 0.0		
	LAB	100.0 ± 0.0	100.0 ± 0.0	100.0 ± 0.0	100.0 ± 0.0		
*T. confusum*	ASC3	92.2 ± 7.7	100.0 ± 0.0	100.0 ± 0.0	100.0 ± 0.0	1.00	0.405
	ASC9	100.0 ± 0.0	99.4 ± 0.6	100.0 ± 0.0	100.0 ± 0.0	1.00	0.405
	ASC14	80.0 ± 5.9 A	88.9 ± 2.0 AB	100.0 ± 0.0 B	100.0 ± 0.0 B	9.689	<0.001
	V1	68.3 ± 10.3 A	93.9 ± 3.7 B	96.1 ± 1.6 B	100.0 ± 0.0 B	6.680	<0.001
	V2	80.6 ± 7.7 A	95.0 ± 2.0 AB	97.7 ± 1.6 B	100.0 ± 0.0 B	4.532	0.009
	P1	75.5 ± 2.5 A	92.2 ± 3.9 B	96.1 ± 1.8 B	100.0 ± 0.0 B	18.245	<0.001
	AGR-01	100.0 ± 0.0	100.0 ± 0.0	100.0 ± 0.0	100.0 ± 0.0		
	AGR-05	100.0 ± 0.0	100.0 ± 0.0	100.0 ± 0.0	100.0 ± 0.0		
	ASC16	99.4 ± 0.6	100.0 ± 0.0	99.4 ± 0.6	100.0 ± 0.0	0.667	0.579
	GA12	28.9 ± 14.5 A	50.0 ± 12.6 BC	80.0 ± 4.7 AB	100.0 ± 0.0 C	10.055	<0.001
	F1	93.3 ± 1.8 A	87.2 ± 2.5 B	100.0 ± 0.0 C	100.0 ± 0.0 C	15.402	<0.001
	F2	83.9 ± 4.2 A	92.7 ± 3.9 AB	99.4 ± 0.6 B	99.4 0.6 B	6.286	0.002
	F3	91.1 ± 2.9 A	94.4 ± 2.9 AB	99.4 ± 0.6 B	100.0 ± 0.0 B	4.035	0.015
	F4	88.9 ± 2.8	92.2 ± 3.2	99.4 ±0.6	100.0 ± 0.0	6.304	0.002
	F5	82.7 ± 2.3 A	91.6 ± 2.8 B	100.0 ± 0.0 C	100.0 ± 0.0 C	19.359	<0.001
	PP1	85.6 ± 2.1	92.7 ± 2.6	100.0 ± 0.0	100.0 ± 0.0	16.635	<0.001
	P2(1)	100.0 ± 0.0	100.0 ± 0.0	100.0 ± 0.0	100.0 ± 0.0		
	P2(2)	100.0 ± 0.0	100.0 ± 0.0	100.0 ± 0.0	100.0 ± 0.0		
	P3	99.4 ± 0.6	100.0 ± 0.0	100.0 ± 0.0	100.0 ± 0.0	1.00	0.405
	P4	99.4 ± 0.6	100.0 ± 0.0	100.0 ± 0.0	100.0 ± 0.0	1.00	0.405
	S1	71.6 ± 4.4 A	86.6 ± 4.1 B	100.0 ± 0.0 C	100.0 ± 0.0 C	19.799	<0.001
	S2	85.6 ± 4.1 C	89.4 ± 2.8 BC	97.2 ± 1.4 AB	100.0 ± 0.0 A	6.640	0.001
	S3	89.4 ± 1.3 A	92.2 ±1.8 A	98.3 ± 1.1 B	100.0 ± 0.0 B	15.008	<0.001
	SKG1	77.2 ± 2.7 A	95.0 ±5.0 B	100.0 ± 0.0 B	100.0 ± 0.0 B	14.302	<0.001
	SKG2	87.2 ± 2.6 A	91.1 ± 3.2 A	99.4 ±0.6 B	100.0 ± 0.0 B	9.029	<0.001
	SK1	73.3 ± 2.8 A	100.0 ± 0.0 B	100.0 ± 0.0 B	100.0 ± 0.0 B	85.333	<0.001
	SK2	85.0 ± 2.6 A	89.4 ±2.1 A	100.0 ± 0.0 B	100.0 ± 0.0 B	20.216	<0.001
	LAB	100.0 ± 0.0	100.0 ± 0.0	100.0 ± 0.0	100.0 ± 0.0		
*C. ferrugineus*	ASC8	98.9 ± 1.1	100.0 ± 0.0	100.0 ± 0.0	100.0 ±0.0	1.000	0.405
	GA2	78.9 ± 5.9 A	80.6 ± 4.3 A	88.3 ± 2.0 AB	100.0 ± 0.0 B	7.098	0.001
	LAB	100.0 ± 0.0	100.0 ± 0.0	100.0 ± 0.0	100.0 ± 0.0		
*T. castaneum*	ASC10	99.4 ± 0.6	100.0 ± 0.0	100.0 ± 0.0	100.0 ± 0.0	1.000	0.405
	ASC11	6.6 ± 2.3 A	10.6 ± 5.9 A	7.2 ± 0.3 A	100.0 ± 0.0 B	91.093	<0.001
	ASC15	100.0 ± 0.0	100.0 ± 0.0	100.0 ± 0.0	100.0 ± 0.0		
	ASC19	100.0 ± 0.0	100.0 ± 0.0	100.0 ± 0.0	100.0 ± 0.0		
	AGR-04	100.0 ± 0.0	100.0 ± 0.0	100.0 ± 0.0	100.0 ± 0.0		
	LAB	100.0 ± 0.0	100.0 ± 0.0	100.0 ± 0.0	100.0 ± 0.0		
*R. dominica*	ASC11	7.7 ± 3.1 A	24.4 ± 15.3 B	28.9 ± 4.2 B	100.0 ± 0.0 C	154.115	<0.001
	GA6	2.2 ± 0.8 A	1.1 ± 1.1 A	20.0 ±1.1 A	100.0 ± 0.0 B	66.768	<0.001
	GA3	1.6 ± 0.8 A	5.6 ± 1.5 A	16.1 ± 4.7 B	100.0 ± 0.0 C	334.790	<0.001
	GA2	84.4 ±5.6 A	100.0 ± 0.0 B	100.0 ± 0.0 B	100.0 ±0.0 B	7.840	<0.001
	AGR-03	3.3 ± 1.1 A	5.0 ± 1.6 A	26.1 ±7.9 B	100.0 ± 0.0 C	122.985	<0.001
	GA12	1.1 ± 0.7 A	1.1 ± 0.7 A	79.4 ± 8.1 B	100.0 ± 0.0 C	157.973	<0.001
	ASC14	6.6 ± 2.2 A	6.1 ± 1.3 A	20.0 ± 5.3 B	100.0 ± 0.0 C	229.681	<0.001
	AGR-01	35.0 ± 7.4 A	88.3 ± 4.7 B	100.0 ± 0.0 B	100.0 ± 0.0 B	49.571	<0.001
	LAB	100.0 ± 0.0	100.0 ±0.0	100.0 ± 0.0	100.0 ± 0.0		

For each population among the concentrations, means followed by the same letter do not differ significantly according to Tukey’s HSD test at *p* < 0.05. Where no letter exist no significant differences were noted.

**Table 4 insects-15-00802-t004:** LC_50_ and LC_99_ (with the confidence intervals) for the mortality response of adults to 50, 100, 200, and 1000 ppm after a 7-day post-exposure period (in all cases, *df* = 34), determined using probit analysis.

Species	Population	LC_50_ (Confidence Interval)	LC_99_ (Confidence Interval)	Y-Intercept	x^2^	*p*	Diagnosis ^a^
*S. granarius*	ASC8	**	**	**	**	**	Susceptible
	LAB	**	**	**	**	****	Susceptible
*O. surinamensis*	ASC10	**	**	**	**	**	Susceptible
	ASC11	100 *	622 *	−5.69	419.3	<0.01	Resistant
	GA1	71.6 *	176.5 *	−11.02	243.7	<0.01	Resistant
	EXTR1	**	**	**	**	**	Susceptible
	LAB	**	**	**	**	****	Susceptible
*S. oryzae*	GA2	0.7 *	27.6 *	0.16	13.8	0.99	Susceptible
	GA1	0.7 *	27.6 *	0.16	13.7	0.99	Susceptible
	ASC7	**	**	**	**	**	Susceptible
	ASC11	34.2 (16–45)	176.1 (119.5–560)	−5.02	113.0	<0.01	Resistant
	U1	0.7 *	27.6 *	0.16	13.7	0.99	Susceptible
	AGR-01	**	**	**	**	**	Susceptible
	LAB	**	**	**	**	****	Susceptible
*T. confusum*	ASC3	**	**	**	**	**	Susceptible
	ASC9	**	**	**	**	**	Susceptible
	ASC14	17.8 (6.3–27)	141.7 (106.4–277)	−3.23	31.3	0.54	Resistant
	V1	24.9 (6.6–36.9)	157.3 (107.7–557.3)	−4.06	88.9	<0.01	Resistant
	V2	20.8 (0.4–32.8)	87.9 (67.8–615.1)	−4.9	50.8	0.03	Resistant
	P1	18.3 (6.7–27.4)	139.9 (105–276)	−3.33	17.3	0.99	Resistant
	AGR-01	**	**	**	**	**	Susceptible
	AGR-05	**	**	**	**	**	Susceptible
	ASC16	0.7 *	27.6 *	0.16	13.8	0.99	Susceptible
	GA12	47.5 (29.8–58.8)	172.7 (118–549.2)	−6.93	200	<0.01	Resistant
	F1	0.7 *	27.6 *	0.16	13.8	0.99	Susceptible
	F2	0.3 *	79.3 *	0.46	60.3	<0.01	Susceptible
	F3	1.8 *	42.3 *	−0.45	25.8	0.84	Susceptible
	F4	9.0 *	64.6 *	−2.61	15.1	0.99	Resistant
	F5	0.9 *	31.8 *	0.94	24.6	0.88	Susceptible
	PP1	0.7 *	27.6 *	0.16	13.7	0.99	Susceptible
	P2(1)	**	**	**	**	**	Susceptible
	P2(2)	**	**	**	**	**	Susceptible
	P3	**	**	**	**	**	Susceptible
	P4	**	**	**	**	**	Susceptible
	S1	9.7 (0.3–21.4)	167.1 (110–763)	−1.86	44.4	0.10	Resistant
	S2	**	**	**	**	**	Susceptible
	S3	13.9 (0–27.1)	75 (59.2–1027)	−3.63	10.7	0.99	Resistant
	SKG1	0.7 *	27.6 *	0.16	13.7	0.99	Susceptible
	SKG2	10.9 *	68.4 *	−3.02	19.9	0.97	Resistant
	SK1	19.2 *	85.1 *	−4.63	13.6	0.99	Resistant
	SK2	7.0 *	58.8 *	−2.15	25.9	0.83	Resistant
	LAB	**	**	**	**	****	Susceptible
*C. ferrugineus*	ASC8	1.8 *	42.3 *	−0.45	25.8	0.84	Susceptible
	GA2	5.0 *	1017 *	−0.71	58.8	<0.01	Resistant
	LAB	**	**	**	**	****	Susceptible
*T. castaneum*	ASC10	**	**	**	**	**	Susceptible
	ASC11	82.6 *	313 *	−7.71	87.4	<0.01	Resistant
	ASC15	**	**	**	**	**	Susceptible
	ASC19	**	**	**	**	**	Susceptible
	AGR-04	**	**	**	**	**	Susceptible
	LAB	**	**	**	**	****	Susceptible
*R. dominica*	ASC11	146 *	2758 *	−3.95	114.7	<0.01	Resistant
	GA6	193.2 *	1517 *	−5.93	89.8	<0.01	Resistant
	GA3	173.5 *	1882 *	−5.03	85.5	<0.01	Resistant
	GA2	22.3 *	90.1 *	−5.18	32.7	0.30	Resistant
	AGR-03	139.9 *	888.8 *	−6.21	55.2	<0.01	Resistant
	GA12	123.6 *	293.5 *	−12.96	289.8	<0.01	Resistant
	ASC14	209.1 *	1607 *	−6.09	66.5	<0.01	Resistant
	AGR-01	**	**	**	**	**	Susceptible
	LAB	**	**	**	**	****	Susceptible

* The confidence intervals could not be estimated precisely. ** The LC value and the resistance ratio could be estimated precisely. ^a^ The diagnosis reported here for the dose–response protocol was the same as the diagnosis determined by FAO protocol, with resistance defined by an LC_50_ value of >2.85, as suggested by Wakil et al. [33].

**Table 5 insects-15-00802-t005:** Mean number (% ± SE) of adults of *S. granarius*, *O. surinamensis*, *S. oryzae*, *T. confusum*, *C. ferrugineus*, *T. castaneum*, and *R. dominica* that were found dead at 50, 100, 200, and 1000 ppm after a 7-day post-exposure period (*df_total_
*= 35).

Species	Population	50	100	200	1000	F	*p*
*S. granarius*	ASC8	100.0 ± 0.0	100.0 ± 0.0	100.0 ± 0.0	100.0 ± 0.0		
	LAB	100.0 ± 0.0	100.0 ± 0.0	100.0 ± 0.0	100.0 ± 0.0		
*O. surinamensis*	ASC10	100.0 ± 0.0	100.0 ± 0.0	100.0 ± 0.0	100.0 ± 0.0		
	ASC11	11.6 ± 4.1	37.2 ± 15.9	96.6 ± 2.2	100.0 ± 0.0	28.002	<0.001
	GA1	10.0 ± 3.8	93.9 ± 3.3	94.4 ±2.8	100.0 ± 0.0		
	EXTR1	100.0 ± 0.0	100.0 ± 0.0	100.0 ± 0.0	100.0 ± 0.0		
	LAB	100.0 ± 0.0	100.0 ± 0.0	100.0 ± 0.0	100.0 ± 0.0		
*S. oryzae*	GA2	100.0 ± 0.0	100.0 ± 0.0	100.0 ± 0.0	100.0 ± 0.0		
	GA1	99.4 ± 0.6	100.0 ± 0.0	100.0 ± 0.0	100.0 ± 0.0	1.000	0.405
	ASC7	100.0 ± 0.0	100.0 ± 0.0	100.0 ± 0.0	100.0 ± 0.0		
	ASC11	65.0 ± 11.8 A	100.0 ± 0.0 B	100.0 ± 0.0 B	100.0 ± 0.0 B	8.733	<0.001
	U1	99.4 ± 0.6	100.0 ± 0.0	100.0 ± 0.0	100.0 ± 0.0	1.000	0.405
	AGR-01	100.0 ± 0.0	100.0 ± 0.0	100.0 ± 0.0	100.0 ± 0.0		
	LAB	100.0 ± 0.0	100.0 ± 0.0	100.0 ± 0.0	100.0 ± 0.0		
*T. confusum*	ASC3	100.0 ± 0.0	100.0 ± 0.0	100.0 ± 0.0	100.0 ± 0.0		
	ASC9	100.0 ± 0.0	100.0 ± 0.0	100.0 ± 0.0	100.0 ± 0.0		
	ASC14	85 ± 4.4	98.3 ± 1.1	100.0 ± 0.0	100.0 ± 0.0		
	V1	77.7 ± 8.9	98.9 ± 0.7	100.0 ± 0.0	100.0 ± 0.0		
	V2	90.6 ± 6.0	99.4 ± 0.6	98.9 ± 0.7	100.0 ± 0.0	2.103	0.119
	P1	85.0 ± 0.0 A	99.4 ± 0.6 B	98.9 ± 1.1 B	100.0 ± 0.0 B	21.208	<0.001
	AGR-01	100.0 ± 0.0	100.0 ± 0.0	100.0 ± 0.0	100.0 ± 0.0		
	AGR-05	100.0 ± 0.0	100.0 ± 0.0	100.0 ± 0.0	100.0 ± 0.0		
	ASC16	99.4 ± 0.6	100.0 ± 0.0	100.0 ± 0.0	100.0 ± 0.0	1.000	0.405
	GA12	41.1 ± 15.5 A	81.1 ± 6.7 B	100.0 ± 0.0 B	100.0 ± 0.0 B	10.710	<0.001
	F1	97.2 ± 1.2	97.2 ± 1.2	100.0 ± 0.0	100.0 ± 0.0	3.509	0.026
	F2	92.7 ± 1.6 A	95.0 ± 1.6 AB	100.0 ± 0.0 B	100.0 ± 0.0 B	4.468	0.010
	F3	93.3 ± 2.8	97.2 ± 1.6	100.0 ± 0.0	100.0 ± 0.0	3.559	0.025
	F4	91.6 ± 2.2	98.9 ± 1.1	100.0 ± 0.0	100.0 ± 0.0	10.582	<0.001
	F5	94.4 ± 1.5	96.1 ± 1.6	100.0 ± 0.0	100.0 ± 0.0		
	PP1	92.7 ± 1.4 A	97.7 ± 1.2 B	100.0 ± 0.0 B	100.0 ± 0.0 B	12.794	<0.001
	P2(1)	100.0 ± 0.0	100.0 ± 0.0	100.0 ± 0.0	100.0 ± 0.0		
	P2(2)	100.0 ± 0.0	100.0 ± 0.0	100.0 ± 0.0	100.0 ± 0.0		
	P3	100.0 ± 0.0	100.0 ± 0.0	100.0 ± 0.0	100.0 ± 0.0		
	P4	100.0 ± 0.0	100.0 ± 0.0	100.0 ± 0.0	100.0 ± 0.0		
	S1	83.3 ± 3.4 A	96.6 ± 2.7 B	100.0 ± 0.0 B	100.0 ± 0.0 B	12.952	<0.001
	S2	96.1 ± 1.1	96.1 ± 1.6	99.4 ± 0.6	100.0 ± 0.0	4.222	0.013
	S3	91.6 ± 1.6 A	97.2 ± 1.2 B	99.4 ± 0.6 B	100.0 ± 0.0 B	12.723	<0.001
	SKG1	88.9 ± 2.1 A	100.0 ± 0.0 B	100.0 ± 0.0 B	100.0 ± 0.0 B	26.230	<0.001
	SKG2	90.0 ± 2.2 A	98.9 ± 0.7 B	99.2 ±0.6 B	100.0 ± 0.0 B	12.766	<0.001
	SK1	86.1 ± 1.8 A	100.0 ± 0.0 B	100.0 ± 0.0 B	100.0 ± 0.0 B	58.140	<0.001
	SK2	93.3 ± 1.6 A	98.9 ± 0.7 B	100.0 ± 0.0 B	100.0 ± 0.0 B	12.279	<0.001
	LAB	100.0 ± 0.0	100.0 ± 0.0	100.0 ± 0.0	100.0 ± 0.0		
*C. ferrugineus*	ASC8	98.9 ± 1.1	100.0 ± 0.0	100.0 ± 0.0	100.0 ± 0.0	1.000	0.405
	GA2	88.4 ± 4.6 A	84.4 ± 2.9 A	93.9 ± 1.3 AB	100.0 ± 0.0 B	5.718	0.003
	LAB	100.0 ± 0.0	100.0 ± 0.0	100.0 ± 0.0	100.0 ± 0.0		
*T. castaneum*	ASC10	99.4 ± 0.6	100.0 ± 0.0	100.0 ± 0.0	100.0 ± 0.0	1.000	0.405
	ASC11	18.3 ± 4.8 A	50.0 ±6.8 B	93.9 ± 2.8 C	100.0 ± 0.0 C	82.535	<0.001
	ASC15	100.0 ± 0.0	100.0 ± 0.0	100.0 ± 0.0	100.0 ± 0.0		
	ASC19	100.0 ± 0.0	100.0 ± 0.0	100.0 ± 0.0	100.0 ± 0.0		
	AGR-04	98.3 ± 1.1	100.0 ± 0.0	100.0 ± 0.0	100.0 ± 0.0	2.000	0.134
	LAB	100.0 ± 0.0	100.0 ± 0.0	100.0 ± 0.0	100.0 ± 0.0		
*R. dominica*	ASC11	32.7 ± 5.8 A	35.0 ± 5.0 A	45.0 ± 4.2 A	100.0 ± 0.0 B	51.940	<0.001
	GA6	15.6 ± 2.9 A	14.4 ± 2.8 A	46.1 ± 7.5 B	100.0 ± 0.0 C	87.637	<0.001
	GA3	20.6 ± 3.3 A	26.6 ± 3.2 AB	43.9 ± 8.1 B	100.0 ± 0.0 C	59.765	<0.001
	GA2	90.6 ± 4.2 A	100.0 ± 0.0 B	100.0 ± 0.0 B	100.0 ± 0.0 B	5.048	0.006
	AGR-03	15.6 ± 2.6 A	21.1 ± 1.1 A	71.1 ± 5.6 B	100.0 ± 0.0 C	161.135	<0.001
	GA12	6.6 ± 2.3 A	6.1 ± 2.0 A	98.8 ± 0.7 B	100.0 ± 0.0 B	1142.341	<0.001
	ASC14	11.6 ± 1.6 A	17.7 ± 3.0 A	37.7 ± 5.7 B	100.0 ± 0.0 C	143.907	<0.001
	AGR-01	100.0 ± 0.0	100.0 ±0.0	100.0 ± 0.0	100.0 ± 0.0		
	LAB	100.0 ± 0.0	100.0 ±0.0	100.0 ± 0.0	100.0 ± 0.0		

For each population among the concentrations, means followed by the same letter do not differ significantly according to Tukey’s HSD test at *p* < 0.05. Where no letter exist no significant differences were noted.

## Data Availability

Data will be made available upon request.

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
