# Peer review of "Assessment of Phosphine Resistance in Major Stored-Product Insects in Greece Using Two Diagnostic Protocols"

_insects, 2024, doi:10.3390/insects15100802_

Round 1

Reviewer 1 Report

Comments and Suggestions for Authors

Comments and Suggestions for Authors

I have read with attention the ms titled: " Assessment of phosphine resistance in major stored-product insects in Greece using two diagnostic protocols", and authored by Paraskevi Agrafioti1,2*, Efstathios Kaloudis3, Dimitrios Kateris1 and Christos G. Athanassiou1,2

Below you will find my remarks, comments, and suggestions. 

Line 32: Mites are not insects. You can refer to them as arthropods instead. Or remove mites related info.

Line 31: Authors mentioned "insects and mites" but later only focused on insect species when discussing resistance. It might be worth elaborating on mite resistance or removing mites from the resistance discussion if not relevant.

Line 39: Replace ‘is’ with ‘are’

Lines 55-59: Instead of repeating 'Coleoptera' multiple times, you could mention it just once at the beginning, like this: Coleopteran such as……….

Line 59. Add period at the end of sentence.

Lines 64-65: The use of "inconsistent and sometimes conflicting results" might benefit from further elaboration on the nature of these inconsistencies or what causes the conflicting results.

Lines 64-65: Please replace "main" with "major storage pests" to avoid repetition.

Line 67: Please remove "a" before "relatively".

Lines 65-69: Please break this sentence “Fundamentally……………………. different research groups” into two sentences for clear understanding.

Line 70: Please remove the repetitive "of exposure".

Line 91: Please mention what was the sample of infested commodities collected from storage facilities.

Lines 95-97: Please remove common name of S, granaries because other species common names are not there.

Line 101: remove ‘reference’ word

Line 111-112: Did you check the insects with sharp items, needles, or forceps to ensure they were truly dead? Insects sometimes pretend to be dead.

Line 116: Please remove the year of the citation and add citation number.

Line 118: Please remove the mention of '3 replicates' as it’s already stated that the experiment was repeated three times.

Line 125: Do you mean growth chambers? If yes, please specify it.

Line 130: Please remove '3 replicates'.

Lines 131-133: Please rephrase or remove the information about phosphine production and measurements, as it was already mentioned in the previous paragraph to avoid redundancy.                                                                                                                                  Lines 107-135: Please combine the details about phosphine production and measurement into a single paragraph for both protocols. You can rephrase it as:

'For both protocols, the whole experiment was repeated three times with new phosphine produced each time. Phosphine production was described in detail by Agrafioti et al. [18], and concentrations were measured using quantitative gas chromatography (GC) with a GC-2010 Plus (Shimadzu, Kyoto, Japan) instrument equipped with a GS-Q column (30 m long x 0.25 mm i.d., 0.25-µm film thickness; MEGA S.r.I., Milan, Italy), as suggested by Sakka and Athanassiou [24]. Additionally, glass tubes from the Draeger Company (Draeger Safety AG & Co., Lubeck, Germany) were used. The protocols were carried out under laboratory conditions set at 25°C and 55% r.h. in chambers.'

Line 139: Please add the letter 't' after 'no' to correct it to 'not'.

Line 140: Please remove ‘Statistical Analysis’

Lines 141-143: What software was used for ANOVA and Tukey’s test?

Lines 222-249: It’s good that you compared your results with other studies, but it would be helpful to mention if the methods they used, like fumigation protocols or exposure times, were similar to yours. That way, the resistance comparisons would be more meaningful.

Line 257: Please write out 'days' instead of 'd' to maintain consistency, as 'days' is used throughout the rest of the manuscript.

Line 260: The statement 'almost all individuals were close to death' need to clarify. Please provide more precise info to describe what this means in terms of observed insect behavior or physiological state?

Lines 276-278: Could R. dominica’s internal feeding behavior be another reason for its lower susceptibility to phosphine?

Comments on the Quality of English Language

The manuscript is generally well-written with clear and understandable language. However, there are a few areas where consistency and clarity could be improved. For instance, some sentences could be rephrased for better flow and precision, especially in the discussion and methodology sections. Addressing these minor issues would further enhance the overall readability of the manuscript.

Author Response

Comments from the reviewers:

Reviewer 1

I have read with attention the ms titled: “Assessment of phosphine resistance in major stored-product insects in Greece using two diagnostic protocols", and authored by Paraskevi Agrafioti1,2*, Efstathios Kaloudis3, Dimitrios Kateris1 and Christos G. Athanassiou1,2

Below you will find my remarks, comments, and suggestions.

REPLY: The authors would like to thank the reviewer for the comments and recommendations on improving the quality of the manuscript before resubmission. A revised version is submitted with the proposed corrections/additions addressed. A point-by-point reply on each comment/correction can be found below.

Line 32: Mites are not insects. You can refer to them as arthropods instead. Or remove mites related info.

REPLY: Correct. We have modified this sentence accordingly. Please see lines 31-32.

Line 31: Authors mentioned "insects and mites" but later only focused on insect species when discussing resistance. It might be worth elaborating on mite resistance or removing mites from the resistance discussion if not relevant.

REPLY: We have changed the sentence accordingly, please see lines 31-32.

Line 39: Replace ‘is’ with ‘are’

REPLY: It is mentioned as “the control”, so we do not need plural form.

Lines 55-59: Instead of repeating 'Coleoptera' multiple times, you could mention it just once at the beginning, like this: Coleopteran such as……….

REPLY: These species are mentioned for the first time throughout the text, so it is necessary to mention the full scientific name for each species, separately.

Line 59. Add period at the end of sentence.

REPLY: Done, please see line 59.

Lines 64-65: The use of "inconsistent and sometimes conflicting results" might benefit from further elaboration on the nature of these inconsistencies or what causes the conflicting results.

REPLY: We have changed this sentence, please see lines 64-65.

Lines 64-65: Please replace "main" with "major storage pests" to avoid repetition.

REPLY: Done, please see line 64.

Line 67: Please remove "a" before "relatively".

REPLY: Revised, please see line 67.

Lines 65-69: Please break this sentence “Fundamentally……………………. different research groups” into two sentences for clear understanding.

REPLY: Revised. Please see lines 65-69.

Line 70: Please remove the repetitive "of exposure".

REPLY: We have deleted the “of exposure”, please see line 70.

Line 91: Please mention what was the sample of infested commodities collected from storage facilities.

REPLY: We have added “at least 100 g” ….. “different cereals”  Please see lines 91-92.

Lines 95-97: Please remove common name of S, granaries because other species common names are not there.

REPLY: We have added the full scientific name of S. granarius, since it was mentioned for the first the through the text.

Line 101: remove ‘reference’ word

REPLY: Deleted, please see line 101.

Line 111-112: Did you check the insects with sharp items, needles, or forceps to ensure they were truly dead? Insects sometimes pretend to be dead.

REPLY: Right! We have added … “and checked ….. a paintbrush”. Please see lines 112-113.

Line 116: Please remove the year of the citation and add citation number.

REPLY: We have added the citation number.

Line 118: Please remove the mention of '3 replicates' as it’s already stated that the experiment was repeated three times.

REPLY: We have modified this phrase as “by 3 sub-replicates”, please see line 118.

Line 125: Do you mean growth chambers? If yes, please specify it.

REPLY: Correct, it is an incubation chamber.

Line 130: Please remove '3 replicates'.

REPLY: We have modified this phrase as “by 3 sub-replicates”, please see line 130.

Lines 131-133: Please rephrase or remove the information about phosphine production and measurements, as it was already mentioned in the previous paragraph to avoid redundancy.

REPLY: Correct, we have deleted the sentence “Phosphine measurements …. As described above”. Please see line 133.

Lines 107-135: Please combine the details about phosphine production and measurement into a single paragraph for both protocols. You can rephrase it as: 'For both protocols, the whole experiment was repeated three times with new phosphine produced each time. Phosphine production was described in detail by Agrafioti et al. [18], and concentrations were measured using quantitative gas chromatography (GC) with a GC-2010 Plus (Shimadzu, Kyoto, Japan) instrument equipped with a GS-Q column (30 m long x 0.25 mm i.d., 0.25-µm film thickness; MEGA S.r.I., Milan, Italy), as suggested by Sakka and Athanassiou [24]. Additionally, glass tubes from the Draeger Company (Draeger Safety AG & Co., Lubeck, Germany) were used. The protocols were carried out under laboratory conditions set at 25°C and 55% r.h. in chambers.'

REPLY: Correct! We have added this paragraph in subsection 2.3 “CORESTA protocol”.

Line 139: Please add the letter 't' after 'no' to correct it to 'not'.

REPLY: Done, see line 139.

Line 140: Please remove ‘Statistical Analysis’

REPLY: Done.

Lines 141-143: What software was used for ANOVA and Tukey’s test?

REPLY: SPSS was used. It was mentioned in line 140.

Lines 222-249: It’s good that you compared your results with other studies, but it would be helpful to mention if the methods they used, like fumigation protocols or exposure times, were similar to yours. That way, the resistance comparisons would be more meaningful.

REPLY: Thank you for your comment. We mentioned it in the third and fifth paragraphs of the discussion section.

Line 257: Please write out 'days' instead of 'd' to maintain consistency, as 'days' is used throughout the rest of the manuscript.

REPLY: Revised, please see line 252.

Line 260: The statement 'almost all individuals were close to death' need to clarify. Please provide more precise info to describe what this means in terms of observed insect behavior or physiological state?

REPLY: We have added this phrase “which means …. affected”.

Lines 276-278: Could R. dominica’s internal feeding behavior be another reason for its lower susceptibility to phosphine?

REPLY: It could be another reason since the lesser grain borer is an internal feeder. However, the commodity sorption can be a major factor in determining whether a lethal concentration of fumigant is achieved under sufficiently airtight conditions. Also, phosphine sorption is known to be influenced by a given commodity's previous fumigation history, moisture content, particle size and composition, exposure period, and dose.

References:

  • Berck, B., Sorption of phosphine by cereal products. J Agric Food Chem 16: 419–425 (1968).
  • Dhaliwal, G.S., Analysis of some important factors affecting sorption of fumigants by food commodities. J Food Sci Technol 12: 1–5 (1975).
  • Reddy, P.V., Rajashekar, Y., Begum, K., Leelaja, B.C., Rajendran, S. The relation between phosphine sorption and terminal gas concentrations in successful fumigation of food commodities. Pest Mang. Sci. 63: 96-103 (2007).

Comments on the Quality of English Language: The manuscript is generally well-written with clear and understandable language. However, there are a few areas where consistency and clarity could be improved. For instance, some sentences could be rephrased for better flow and precision, especially in the discussion and methodology sections. Addressing these minor issues would further enhance the overall readability of the manuscript.

REPLY: Thank you for these comments. We have addressed/incorporated the minor issues throughout the text.

Reviewer 2 Report

Comments and Suggestions for Authors

These are my main comments on the manuscript (Insects-3220325) entitled “Assessment of phosphine resistance in major stored-product insects in Greece using two diagnostic protocols”. This work investigates the phosphine resistance on pests of stored grains. Following substantial revisions should be incorporated in the manuscript prior to acceptance.

L.15: Change “Dose Response or dose response” by “dose-response”. Please, correct in all manuscript.

Ls.15-16: Authors should briefly explain the two protocols used in this study.

Ls.16-17. Which insect populations? Any species?

Ls.17-18: More information about results obtained in this study is needed.

L.24: Keywords should be in alphabetic order. Also, keywords serve to widen the opportunity to be retrieved from a database. To put words that already are into title and abstracts makes KW not useful. Please choose terms that are neither in the title nor in abstract.

Ls.38-40: Provide references that support the use of insecticides.

Ls.40-42: Rewrite this sentence.

L.49: ...many pests around...

L.52: Change “(F.)” by “Fabricius”.

L.56: Again, change “(L.)” by “Linnaeus”.

L.59: Place “.” after Resistance.

Ls.59-61: Revise this sentence to eliminate rewordiness.

Ls.69-70: again, revise this sentence to eliminate rewordiness.

L.75: Change “(F.)” by “Fabricius”.

L.81: …of L. serricorne, the…

L.84: Also, a hypothesis for this study is needed.

Ls.85-87: The objective is not clear, please rewrite.

Ls.95-97: More details about insect populations of this species are needed.

Ls.105-106: Delete this figure. Authors can provide this information in one sentence.

L.110: Change “1 lt” by “1 L”.

L.118: Delete “(3 replicates)”.

Ls.120-122: More information about chromatography protocol use is need.

L.130: L.118: Delete “(3 replicates)”.

L.143 Define HSD.

Ls.152-160: Information on insect populations must be previously mentioned in the materials and methods section.

L.153: … at 43 ppm…

Ls.157-158: Survival rates? Or…Number of insect dead? Verify in all manuscript.

Ls.161-162: Rewrite the table legend.

Table 1. Also, provide the slope for each probit analysis.

Ls.184-185: Rewrite the table legend.

Ls204-220: This result should be previously mentioned in the materials and methods section.

L.264: Change “L.” by “Linnaeus”.

L.294: Any conclusions? You have several results obtained (experiments) in your research but you have not concluded anything. Also, explain in more details how your findings are important for controlling pests stored grains (see abstract, Ls.21-23) since they you just looked at the phosphine toxicity using two diagnostic protocols.

Comments on the Quality of English Language

Moderate editing of English language required.

Author Response

Reviewer 2

These are my main comments on the manuscript (Insects-3220325) entitled “Assessment of phosphine resistance in major stored-product insects in Greece using two diagnostic protocols”. This work investigates the phosphine resistance on pests of stored grains. Following substantial revisions should be incorporated in the manuscript prior to acceptance.

REPLY: The authors would like to thank the reviewer for the comments and recommendations on improving the quality of the manuscript before resubmission. A revised version is submitted with the proposed corrections/additions addressed. A point-by-point reply on each comment/correction can be found below.

L.15: Change “Dose Response or dose response” by “dose-response”. Please, correct in all manuscript.

REPLY: Revised.

Ls.15-16: Authors should briefly explain the two protocols used in this study.

REPLY: We have revised this sentence, please see lines 15-16.

Ls.16-17. Which insect populations? Any species?

REPLY: We have made some modifications to this sentence.

Ls.17-18: More information about results obtained in this study is needed.

REPLY: Indeed. We added “Specifically …..  susceptible to phosphine”.

L.24: Keywords should be in alphabetic order. Also, keywords serve to widen the opportunity to be retrieved from a database. To put words that already are into title and abstracts makes KW not useful. Please choose terms that are neither in the title nor in abstract.

REPLY: We have modified the keywords.

Ls.38-40: Provide references that support the use of insecticides.

REPLY: Done.

Ls.40-42: Rewrite this sentence.

REPLY: Revised, see lines 40-42.

L.49: ...many pests around...

REPLY: Revised.

L.52: Change “(F.)” by “Fabricius”.

REPLY: The correct scientific format writing name is: Rhyzopertha dominica (F.) (Coleoptera: Bostrychidae).

L.56: Again, change “(L.)” by “Linnaeus”.

REPLY: The correct scientific format writing name is: Sitophilus oryzae (L.) (Coleoptera: Curculionidae)

L.59: Place “.” after Resistance.

REPLY: Done.

Ls.59-61: Revise this sentence to eliminate rewordiness.

REPLY: Done.

Ls.69-70: again, revise this sentence to eliminate rewordiness.

REPLY: Done.

L.75: Change “(F.)” by “Fabricius”.

REPLY: The correct scientific format writing name is: Lasioderma serricorne (F.) (Coleoptera: Anobiidae)

L.81: …of L. serricorne, the…

REPLY: Done.

L.84: Also, a hypothesis for this study is needed.

REPLY: Correct, we have modified it.

Ls.85-87: The objective is not clear, please rewrite.

REPLY: We made some revisions in the objective part.

Ls.95-97: More details about insect populations of this species are needed.

REPLY: Done, please see the Table 1.

Ls.105-106: Delete this figure. Authors can provide this information in one sentence.

REPLY: Done.

L.110: Change “1 lt” by “1 L”.

REPLY: Done.

L.118: Delete “(3 replicates)”.

REPLY: We have added a short paragraph explaining the phosphine measurements.

Ls.120-122: More information about chromatography protocol use is need.

REPLY: More details about the chromatography procedure are provided by Sakka and Athanassiou [24].

  • Sakka, M.K.; Athanassiou, C.G. Evaluation of phosphine resistance in three Sitophilus species of different geographical origins using two diagnostic protocols. Agriculture. 2023, 13, 1068. https://doi.org/10.3390/agriculture13051068

L.130: L.118: Delete “(3 replicates)”.

REPLY: Please see the above comment.

L.143 Define HSD.

REPLY: Done.

Ls.152-160: Information on insect populations must be previously mentioned in the materials and methods section.

REPLY: In these lines, we mentioned the results of the “Dose-Response” protocol.

L.153: … at 43 ppm…

REPLY: Revised.

Ls.157-158: Survival rates? Or…Number of insect dead? Verify in all manuscript.

REPLY: In this sentence, we are referring to the survival rates.

Ls.161-162: Rewrite the table legend.

REPLY: Done.

Table 1. Also, provide the slope for each probit analysis.

REPLY: Recent published papers have not been included the slope of the probit analysis, anymore. Thus for this reason, we would prefer to maintain the table as is.

Some references are provided below:

  • Athanassiou C.G., Phillios, T.W., Arthur, F.H., Aikins, M.J., Agrafioti, P., Hartzer, K.L. (2020). Efficacy of phosphine fumigation for different life stages of Trogoderma inclusum and Dermestes maculatus (Coleptera: Dermestidae). Journal of Stored Products Research 86, 101556.
  • Sakka M., Athanassiou C.G. (2021). Population-Mediated responses of Lasioderma serricorne (Coleoptera: Anobiidae) to different diagnostic protocols for phosphine efficacy. Journal of Economic Entomology 114, 885-890.
  • Lampiri, E., Athanassiou C.G. (2021). Insecticidal effect of phosphine on eggs of the khapra beetle (Coleoptera: Dermestidae). Journal of Economic Entomology 114, 1389-1400.
  • Sakka M.K., Nakas C.T., Bochtis, D., Athanassiou C.G. (2023). Quick knockdown results in high mortality: is this theory correct? A case study with phosphine and the red flour beetle. Pest Management Science 79, 3740-3748.
  • Sakka M., Athanassiou, C.G. (2023). Evaluation of phosphine resistance in three Sitophilus species of different geographical origins using two diagnostic protocols. Agriculture 13, 1068.

Ls.184-185: Rewrite the table legend.

REPLY: Revised.

Ls204-220: This result should be previously mentioned in the materials and methods section.

REPLY: Thank you for this comment. In these lines, Figure 1 is analyzed, in which the two diagnostic protocols used in the manuscript are illustrated in Greek storage facilities.

L.264: Change “L.” by “Linnaeus”.

REPLY: The correct scientific format writing name is: Tenebrio molitor (L. (Coleoptera: Tenebrionidae).

L.294: Any conclusions? You have several results obtained (experiments) in your research but you have not concluded anything. Also, explain in more details how your findings are important for controlling pests stored grains (see abstract, Ls.21-23) since they you just looked at the phosphine toxicity using two diagnostic protocols.

REPLY: We added a paragraph in the discussion section.

Comments on the Quality of English Language:Moderate editing of English language required.

REPLY: Thank you for the comment. We have dome some changes through the text.

Reviewer 3 Report

Comments and Suggestions for Authors

See the review report for detailed comments

Author Response

Reviewer 3

Ms No: insects-3220325 Assessment of phosphine resistance in major stored-product insects in Greece using two diagnostic protocols

Reviewer’s remarks

Abstract: Lines 15-17: Two assessment protocols, Dose-Response and CORESTA, are used to estimate phosphine resistance. The results show that 13.3% of the field populations are resistant, with mortality rates increasing with higher phosphine concentrations

REPLY: The authors would like to thank the reviewer for the comments and recommendations on improving the quality of the manuscript before resubmission. A revised version is submitted with the proposed corrections/additions addressed. A point-by-point reply on each comment/correction can be found below.

Remark: This claim is not fully supported by the Results obtained in this study; The statement pertains only to Dose – Response method; whereas testing by CORESTA revealed that all the populations were found to be susceptible

REPLY: Correct. We have dome some modifications in the abstract.

As stated in line numbers 74-77, the CORESTA protocol based on higher concentrations and longer exposure intervals for the cigarette or tobacco beetle, Lasioderma serricorne (F.) (Coleoptera: Anobiidae) and the tobacco moth, Ephestia elutella (Hübner) (Lepidoptera: Pyralidae) [31] was used to compensate and mitigate failures due to phosphine resistance; It’s a mitigation strategy for managing the resistance levels in the said insect species. What is the rationale for using this protocol as a diagnostic test?

REPLY: The Cooperation Center for Scientific Research to Tobacco (CORESTA) bioassay is based on exposures of 4 days at 200 ppm, and if there are surviving individuals when the tobacco temperature is >20oC, for the susceptible populations, while for the resistant population 700 ppm for 10 days of exposure. In our case, we used the concentration of 300 ppm for 6 days of exposure and we found that all tested populations were found to be susceptible to phosphine, under laboratory conditions (temperature and relative humidity).

A diagnostic test should able to detect early, the resistance development in insects. The concentrations and exposure time used in CORESTA protocol are sufficiently higher and longer respectively that it can not detect the resistance development.

REPLY: Correct! The CORESTA protocol is focused only on two species, the tobacco beetle and the tobacco moth. We would like to evaluate the Greek stored product insect populations by using two diagnostic protocols under laboratory conditions (temperature and relative humidity) to check the susceptibility to phosphine.

Lines 270-272, state …… all stored product species were classified as susceptible to phosphine since complete mortality was observed. “”

REPLY: Done.

Remark : Unlike insecticides, toxicity of the fumigant is estimated in terms of cT values and a combination of high toxicity and longer exposure period obviously results in higher toxicity of phosphine to these populations. Hence, there is a need for major revision of interpretations of the results. The dose response protocol estimates the resistance levels of the populations and the CORESTA protocol can be used for the mitigation of phosphine resistance in insect populations. The MS cannot be accepted in its present form and major revision is needed.

REPLY: The authors would like to thank the reviewer for this comment.

Reviewer 4 Report

Comments and Suggestions for Authors

General comments: This study provides valuable insights into the phosphine resistance of stored product insects across various regions of Greece. The methodology employed is scientifically robust, and the results contribute to our understanding of insect resistance to phosphine.

However, the manuscript requires substantial revisions before it can be considered for publication. Specifically, the English language needs improvement. Additionally, the presentation of the scientific data should be more logical and coherent.  

Specific comments.

Line 51: ‘America’ represents South America or North America? Please provide clarity.

Line 82: Please use a proper degree sign for the Unit.

Line 83: Please remove the unnecessary number ‘19’.

Line 91: The authors have mentioned, ‘samples of infested commodities’, it would better help the readers if they could list the commodities sampled, as well.

Line 102: There is a space between the number and the degree C; Other locations (at line 82) do not have space between them. Please maintain consistency.

Line 108: Is it ‘also known as’?

Line 109: Does the samples represent the infested commodity. If yes, they were also fumigated?

Line 151: I think, LC50 values <2.85 ppm are characterized as susceptible.

Table 1: What does those alpha-numeric codes represent? Please provide the information clearly. Please include the details like source commodity and the storage facility, from which the species have been extracted.

Line 173: I am confused. How can the resistance percent increase in post-exposure period?

Line 175: Again, when we don’t know, what is ASC11, its hard to follow.

Line 185: 7 day exposure period?

Table 5: When its all 100%, table 5 seems unnecessary.

Figure 2: When the bars represent the number of insect species, its better to provide the values. By comparing different bar heights its difficult to understand the number.

Line 213: While explaining different regions, it would be clear for readers of other countries, if you could show the regions in the map too.

Overall, the authors have presented the results. But, what about the conclusion to the study? Based on the findings of the study, which method do the authors think is best for phosphine resistance determination. Where can the findings of the study be used?

Comments on the Quality of English Language

English language needs improvement

Author Response

Reviewer 4

General comments: This study provides valuable insights into the phosphine resistance of stored product insects across various regions of Greece. The methodology employed is scientifically robust, and the results contribute to our understanding of insect resistance to phosphine. However, the manuscript requires substantial revisions before it can be considered for publication. Specifically, the English language needs improvement. Additionally, the presentation of the scientific data should be more logical and coherent.

REPLY: The authors would like to thank the reviewer for the comments and recommendations on improving the quality of the manuscript before resubmission. A revised version is submitted with the proposed corrections/additions addressed. A point-by-point reply on each comment/correction can be found below. Regarding the English language, we have made several modifications through the manuscript.

Specific comments.

Line 51: ‘America’ represents South America or North America? Please provide clarity.

REPLY: Revised.

Line 82: Please use a proper degree sign for the Unit.

REPLY: Revised.

Line 83: Please remove the unnecessary number ‘19’.

REPLY: Done.

Line 91: The authors have mentioned, ‘samples of infested commodities’, it would better help the readers if they could list the commodities sampled, as well.

REPLY: We added the Table 1.

Line 102: There is a space between the number and the degree C; Other locations (at line 82) do not have space between them. Please maintain consistency.

REPLY: Revised.

Line 108: Is it ‘also known as’?

REPLY: Correct, we have revised this phrase.

Line 109: Does the samples represent the infested commodity. If yes, they were also fumigated?

REPLY: These samples were wild strains, which means that most of them were also fumigated.

Line 151: I think, LC50 values <2.85 ppm are characterized as susceptible.

REPLY: As suggested by Wakil et al., 2021, this population was characterized as resistant.

  • Wakil, W.; Kavallieratos, N.G.; Usman, M.; Gulzar, S.; El-Shafie, HAF. Detection of Phosphine Resistance in Field Populations of 368 Four Key Stored-Grain Insect Pests in Pakistan. Insects. 2021, 12, 288.

Table 1: What does those alpha-numeric codes represent? Please provide the information clearly. Please include the details like source commodity and the storage facility, from which the species have been extracted.

REPLY: We have added Table 1, including the sampling information. Every population has a unique code.

Line 173: I am confused. How can the resistance percent increase in post-exposure period?

REPLY: In post-exposure period, some of the populations tested were recovered.

Line 175: Again, when we don’t know, what is ASC11, its hard to follow.

REPLY: It is a unique code for each population-species.

Line 185: 7 day exposure period?

REPLY: After 7 days of post-exposure period, which means that we kept those individuals in controlled conditions after the termination of each fumigation protocol, and after 7 days the mortality rates were recorded.

Table 5: When its all 100%, table 5 seems unnecessary.

REPLY: Done.

Figure 2: When the bars represent the number of insect species, its better to provide the values. By comparing different bar heights its difficult to understand the number.

REPLY: Done. We revised the map, please see figure 1.

Line 213: While explaining different regions, it would be clear for readers of other countries, if you could show the regions in the map too.

REPLY: Done.

Overall, the authors have presented the results. But, what about the conclusion to the study? Based on the findings of the study, which method do the authors think is best for phosphine resistance determination. Where can the findings of the study be used?

REPLY: We have added a paragraph in the discussion section.

Round 2

Reviewer 2 Report

Comments and Suggestions for Authors

The manuscript “Assessment of phosphine resistance in major stored-product insects in Greece using two diagnostic protocols” has been improved and all my questions were taken into account. I recommend the publication in “Insects”. 

Comments on the Quality of English Language

Minor editing of English language required.

Author Response

The authors would like to thank the reviewer. 

Reviewer 3 Report

Comments and Suggestions for Authors

The authors have addressed the concerns raised duing the review.

The revised version can be accepted for publication

Author Response

(The authors gave the same response as above.)

Reviewer 4 Report

Comments and Suggestions for Authors

Dear Authors, Thank you for submitting the revised version of your manuscript and addressing the comments and suggestions provided in the previous review. 

Upon careful review of the revised manuscript, I appreciate the efforts you have made to incorporate the feedback. However, I do have the following comments. 

Line 185: Wakil et al., 2021 identified an LC50 value of 2.85 ppm for laboratory susceptible strains. If the LC50 value is less than 2.85, the insects can be categorized as susceptible, and if it is greater than 2.85, they can be categorized as resistant. Based on this, the authors have correctly identified the susceptibility in Table 2, marking a value of 0.7 as susceptible and 102.5 as resistant. However, the statement in line 185 should specify ">2.85" for categorizing insects as resistant, not "<2.85".

Line 221: I realize my previous comments may not have been clear. The statement, ".... ppm after a 7 of post-exposure period", should be written as" 7 days of post-exposure period". 

Figure 1: Although the authors have provided additional information, the clarity of the image is poor, and the numbers below the bar chart are unreadable. Please improve the resolution or formatting to ensure they are clearly visible.

Comments on the Quality of English Language

The English language is sufficiently clear. 

Author Response

Upon careful review of the revised manuscript, I appreciate the efforts you have made to incorporate the feedback. However, I do have the following comments. 

Line 185: Wakil et al., 2021 identified an LC50 value of 2.85 ppm for laboratory susceptible strains. If the LC50 value is less than 2.85, the insects can be categorized as susceptible, and if it is greater than 2.85, they can be categorized as resistant. Based on this, the authors have correctly identified the susceptibility in Table 2, marking a value of 0.7 as susceptible and 102.5 as resistant. However, the statement in line 185 should specify ">2.85" for categorizing insects as resistant, not "<2.85".

REPLY: Revised accordingly.

Line 221: I realize my previous comments may not have been clear. The statement, ".... ppm after a 7 of post-exposure period", should be written as" 7 days of post-exposure period". 

REPLY: Revised.

Figure 1: Although the authors have provided additional information, the clarity of the image is poor, and the numbers below the bar chart are unreadable. Please improve the resolution or formatting to ensure they are clearly visible.

REPLY: Revised.